# Porcine Epidemic Diarrhea Virus Replication in Human Intestinal Cells Reveals Potential Susceptibility to Cross-Species Infection

**DOI:** 10.3390/v15040956

**Published:** 2023-04-13

**Authors:** Zheng Niu, Shujuan Zhang, Shasha Xu, Jing Wang, Siying Wang, Xia Hu, Li Zhang, Lixin Ren, Jingyi Zhang, Xiangyang Liu, Yang Zhou, Liu Yang, Zhenhui Song

**Affiliations:** 1College of Veterinary Medicine, Southwest University, Chongqing 402460, China; nz0511@126.com (Z.N.);; 2College of Veterinary Medicine, Northwest A&F University, Xianyang 712100, China; 3College of Veterinary Medicine, Xinjiang Agricultural University, Ürümqi 830052, China; 4National Center of Technology Innovation for Pigs, Chongqing 402460, China; 5Immunology Research Center, Medical Research Institute, Southwest University, Chongqing 402460, China

**Keywords:** PEDV, human small intestinal epithelial cells, cross-species transmission

## Abstract

Various coronaviruses have emerged as a result of cross-species transmission among humans and domestic animals. Porcine epidemic diarrhea virus (PEDV; family Coronaviridae, genus Alphacoronavirus) causes acute diarrhea, vomiting, dehydration, and high mortality in neonatal piglets. Porcine small intestinal epithelial cells (IPEC-J2 cells) can be used as target cells for PEDV infection. However, the origin of PEDV in pigs, the host range, and cross-species infection of PEDV remain unclear. To determine whether PEDV has the ability to infect human cells in vitro, human small intestinal epithelial cells (FHs 74 Int cells) were inoculated with PEDV LJX and PEDV CV777 strains. The results indicated that PEDV LJX, but not PEDV CV777, could infect FHs 74 Int cells. Furthermore, we observed M gene mRNA transcripts and N protein expression in infected FHs 74 Int cells. A one-step growth curve showed that the highest viral titer of PEDV occurred at 12 h post infection. Viral particles in vacuoles were observed in FHs 74 Int cells at 24 h post infection. The results proved that human small intestinal epithelial cells are susceptible to PEDV infection, suggesting the possibility of cross-species transmission of PEDV.

## 1. Introduction

Coronaviruses (CoVs) cause respiratory and digestive disease in humans and other animals and are responsible for several emerging diseases [1,2]. The severe acute respiratory syndrome (SARS) outbreak in 2002–2003 resulted in 8422 human cases and 916 deaths in 33 countries [3]. In 2012, Middle East respiratory syndrome (MERS) emerged, and over time has resulted in over 2500 human cases and 866 deaths in 27 countries [4]. As of April, 2023, the novel coronavirus disease 2019 (COVID-19) pandemic has resulted in 6.8 million human deaths and 761.4 million cases in 221 countries and territories [5]. Other animals have also been affected by these and other emerging coronaviruses, all of which resulted from cross-species transmission, thus demonstrating the serious threat coronaviruses can pose to humans and other animals globally [6,7]. 

There are six porcine coronaviruses: four Alphacoronaviruses, transmissible gastroenteritis virus (TGEV), porcine respiratory coronavirus (PRCoV), porcine epidemic diarrhea virus (PEDV), and swine acute diarrhea syndrome coronavirus (SADS-CoV); one Betacoronavirus, porcine hemagglutinating encephalomyelitis virus (PHEV); and one Deltacoronavirus, porcine deltacoronavirus (PDCoV). TGEV, PEDV, SADS-CoV, and PDCoV cause severe enteritis that is fatal in piglets, PHEV causes digestive and/or neurological disease, and PRCoV causes a mild respiratory disease [8]. The recent and rapid global dissemination of highly pathogenic variants of PEDV and PDCoV highlights the critical health threat associated with newly emerged swine coronaviruses, and it demonstrates the need for increased resources to understand the virus and its pathogenic potential in mammals.

PEDV emerged in the 1970s in Europe and subsequently spread throughout Asia, likely from bat CoVs; then the virus was introduced into North America in 2013 [9,10]. A serological study indicated that PEDV subsequently spilled over from domestic to feral swine populations in the US [11]. PEDV, a member of the genus Alphacoronavirus in the family Coronaviridae, causes acute diarrhea and/or vomiting, dehydration, and mortality in neonatal piglets [12]. The main PEDV transmission route is fecal–oral; however, airborne transmission via the fecal–nasal route might play a role in pig-to-pig and farm-to-farm transmission [13,14].

During acute PEDV infection in nursing pigs, PEDV initially infects mainly the jejunum and ileum, with the villous enterocytes of the intestine being frequently infected [15,16]. A previous study showed that PEDV recognizes human membrane alanyl aminopeptidase (APN) and infects human cells, including Huh-7 (human liver) and MRC-5 (human lung) cells [17]. However, it is not clear whether PEDV can infect human small intestinal epithelial cells. Porcine small intestinal epithelial cells are the target cells for PEDV infection, and determining whether PEDV can infect human small intestinal epithelial cells is important to reveal the potential risk of PEDV cross-species infection in humans. To address this problem, the present study demonstrated that PEDV is capable of infecting human small intestinal epithelial cells, thus highlighting the risk of cross-species transmission.

## 2. Materials and Methods

### 2.1. Cells, Viruses, and Reagents

Human small intestinal epithelial cells (FHs 74 Int cells) were purchased from Qingqi Biotechnology Development Co., Ltd. (Shanghai, China). African green monkey kidney cells (Vero) and porcine small intestinal epithelial cells (IPEC-J2) were preserved in our laboratory. IPEC-J2, Vero, and FHs 74 Int cells were cultured at 37 °C and 5% CO_2_ in Dulbecco’s modified Eagle’s medium (DMEM) containing 10% fetal bovine serum or in DMEM free of sodium acetonate (Gibco, Grand Island, NY, USA). The PEDV LJX strain and anti-PEDV N antibody were gifted by Dr. YuGuang Fu, Lanzhou Veterinary Research Institute, Chinese Academy of Agricultural Sciences. The PEDV CV777 strain was stored in our laboratory. Anti-β-actin antibodies and goat anti-mouse horseradish peroxidase (HRP)-IgG antibodies were purchased from Proteintech (Rosemont, IL, USA). Agarose was purchased from the Biowest company (Nuaillé, France). Closed protein dry powder was purchased from Boster Biological Technology Co., Ltd. (Wuhan, China). A polyvinylidene fluoride (PVDF) membrane was purchased from Millipore Corporation (Billerica, MA, USA).

### 2.2. Reverse Transcription PCR

IPEC-J2 cells and FHs Int cells were seeded into six-well dishes, and after reaching 90% confluence, they were infected with the PEDV LJX strain and PEDV CV777 strain separately. At 48 h after infection, total RNA was obtained using the TRIzol method. The acquired RNA was reverse transcribed to cDNA using the following reaction mixture: 2 μL of 5 × PrimeScript RT Master Mix (Perfect Real Time) (Takara, Shiga, Japan), 1 μL of total RNA, and 7 μL of RNase Free H_2_O. The reaction conditions were: 37 °C for 15 min, 85 °C for 5 s, and 4 °C for 5 min. The M gene was then amplified using PEDV M gene-specific primers, as shown in Table 1, and the reaction system comprised 12.5 μL of Premix Taq (Takara, RR901), 8.5 μL of RNase Free H_2_O, 1 μL each of upstream and downstream primers, and 2 μL of cDNA. The reaction system was pre-denatured at 94 °C for 3 min. The rest of the reaction conditions comprised 30 cycles of 94 °C for 30 s, 60 °C for 30 s, and 72 °C for 1 min; then there was a final extension at 72 °C for 5 min. The PCR products were separated using 1% agarose gel electrophoresis, observed using a gel-imaging system, and the results were recorded.

### 2.3. Extraction of Cellular Proteins

IPEC-J2 cells or FHs 74 Int cells were evenly inoculated into six-well plates. We set up 12, 24, 36, and 48 h experimental groups with three duplicates of each, and we set up parallel time control groups. When the cells reached 90% confluence, they were inoculated with PEDV-LJX virus solution (multiplicity of infection (MOI) = 0.1). After incubation, the virus fluid was discarded and replaced with DMEM or DMEM without sodium acetonate. The following steps were taken to extract total protein. After gently washing the cells with phosphate buffered saline (PBS) three times, 200 μL of Western and immunoprecipitation (IP) cell cleavage solution containing phosphatase and protease inhibitors was added to each well and incubated for 30 min on ice with vortexing every 10 min. After cleavage, the cells were centrifuged at 4 °C and 12,000 rpm for 15 min, and the supernatant was retained. The protein concentrations in the samples were determined using the bicinchoninic acid (BCA) method. Then, one fifth of the volume of 6 × loading buffer was added to the samples, followed by boiling for 10 min, cooling, and storage at −40 °C.

### 2.4. Western Blotting

The same amounts of protein extracts were subjected to sodium dodecyl sulfate-polyacrylamide gel electrophoresis (SDS-PAGE). The separated proteins were then transferred to the PVDF membrane, followed by blocking for 90 min in 5% skim milk powder in Tris-buffered saline-Tween 20 (TBST). Then, the anti-PEDV N and anti-β-actin antibodies were added and incubated for 15 h at 4 °C. The membranes were then washed 10 times with 1 × TBST buffer for 4 min each time, followed by incubation in HRP-bound goat antibodies for 60 min at room temperature. The FX5 Imaging System (VILBER, Marne-la-Vallée, France) was used to obtain the blot image, and ImageJ (NIH, Bethesda, MD, USA) was used to analyze the gray value of each immunoreactive protein band.

### 2.5. IFA Analyses

Cell climbing slices (Solarbio, Beijing, China; YA0350) were placed at the bottom of the wells of a 24-well plate before seeding with IPEC-J2 cells. PEDV LJX infection, control, and PEDV CV777 infection groups were set up with three repeat wells for each group. The slices were washed with PBS thrice for 5 min each, followed by fixing in 4% paraformaldehyde for 30 min. The slices were again washed with PBS thrice for 5 min each, and then they were blocked with 5% bovine serum albumin (BSA) for 60 min. They were then washed with PBS three times for 3 min each and incubated with the primary antibodies in a wet dish at 4 °C overnight. After washing with PBS thrice for 3 min each, the cells were incubated with anti-rabbit IgG cyanine 3 (Cy3) Fragment antibody (Cell Signaling Technology, Danvers, MA, USA) for 30 min, and then washed three times. Staining was performed using 4′,6-diamidino-2-phenylindole (DAPI) (Beyotime, Jiangsu, China) for 5 min, followed by three PBS washes for 5 min each. Lastly, 3–5 drops of antifade mounting medium (Beyotime) were added, and the samples were visualized using laser confocal microscopy (Axio-Imager LSM-800, Carl Zeiss AG, Oberkochen, Germany).

### 2.6. PEDV S Gene Homology and Evolutionary Analysis

The RNA extraction and reverse transcription methods are shown in Section 2.2. The obtained cDNA was amplified using the primers shown in Table 2 to obtain the full length of the PEDV S gene, and the DNA contained in the band was recovered using SanPrep (Sangon Biotech, Shanghai, China) after agarose electrophoresis. The recovered products were then sent to Sangon Biotech for sequencing. The sequencing results were compared with the PEDV SX strain (KY420075.1) and PEDV CV777 strain (AF353511.1) S genes registered in GenBank using DNAStar 7.0 (DNAStar, Madison, WI, USA). MEGA11 and EvolView [18,19] softwares were then used to construct evolutionary trees of our amplified sequences together with sequences from strains endemic in China, the United States, Japan, South Korea, and other countries.

### 2.7. Absolute Quantitative PCR

To detect virus replication, the relationship between copy number (X) and cycle threshold (Ct) (Y) was established based on the amplification efficiency of the PEDV membrane (M) gene in the PCR instrument. The primers for PEDV M (Table 3) were used to quantify the number of copies of PEDV M. The PEDV M gene plasmid was preserved in our laboratory. The Power SYBR Green PCR Master Mix (Takara, Dalian, China) was used to carry out the PCR reactions according to the manufacturer’s instructions. We used GraphPad Prism 6 software (GraphPad Inc., La Jolla, CA, USA) to analyze the data based on the cycle δCt method. RNA from collected cell samples was extracted using RNAiso plus (Invitrogen, Waltham, MA, USA). Then, 5 × PrimeScript RT Master Mix (Promega, Madison, WI, USA) and the total viral RNA were used to generate cDNA. The M gene was amplified using quantitative real-time polymerase chain reaction (qPCR) in a reaction comprising 10 μL of SYBR PreMix ExTaq II (Takara), 0.5 μL of the forward primer, 0.5 μL of the reverse primer, 2 μL of the cDNA template, and 5 μL of H_2_O. The reaction was preprocessed for 30 s at 95 °C, followed by 40 cycles of 95 °C for 5 s and 64 °C for 30 s. For each sample, the process was repeated three times. Data analysis was based on the Ct measurements. The relative expression levels of PEDV M mRNA were then calculated.

### 2.8. Electron Microscope

FHs 74 Int cells were inoculated into a 60 mm petri dish, and the PEDV-LJX strain (MOI = 0.1) was added when the cells reached about 90% confluence. The cells were sampled at 24 h and 48 h, washed twice, added with 1 mL of DMEM without sodium acetonate, collected from the centrifugation tube using a cell scraper, and centrifuged at 4 °C and 1000 rpm for 5 min. The supernatant was discarded and diluted fixation liquid (3% glutaraldehyde: 10α = 1:5) was added slowly to gently resuspend the cells, which were then left at 4 °C for 5 min. Finally, the samples were centrifuged at 4 °C and 12,000 rpm for 10 min, and the cell pellet was gently and slowly resuspended in 3% glutaraldehyde fixation liquid and stored 4 °C before electron microscopy observation.

### 2.9. Median Tissue Culture Infectious Dose (TCID_50_) Assay of the Virus Titer

FHs 74 Int cells in cell bottles were inoculated with 10^5^ TCID_50_/mL of the PEDV-LJX strain, and a blank control group was also set up. The TCID_50_ values of each group were measured using the Reed–Muench method in Vero cells [20].

### 2.10. Statistical Analysis

All statistical analyses were performed using GraphPad Prism 8.0. All data are presented as the mean ± SD or with the standard errors of the mean (SE) from three independent experiments. One-way analysis of variance (ANOVA) and t-tests were used to determine the statistical differences among multiple groups. *p*-values less than 0.05 were considered statistically significant (in the figures, * *p*-value < 0.05; ** *p*-value < 0.01; *** *p*-value < 0.001; and **** *p*-value < 0.0001).

## 3. Results

### 3.1. PEDV LJX Can Infect FHs 74 Int Cells, but PEDV CV777 Cannot

To investigate whether both PEDV-LJX and PEDV-CV777 could infect FHs 74 Int cells, we detected M gene mRNA transcripts and N gene protein levels using RT-PCR and western blotting, respectively, in FHs 74 Int cells infected with both PEDV strains. As shown in Figure 1A, FHs 74 Int cells inoculated with PEDV-LJX demonstrated the expression of the PEDV N protein, while no PEDV N protein was detected in FHs 74 Int cells inoculated with PEDV-CV777. As shown in Figure 1B, the sample inoculated with PEDV-LJX showed an amplification product at 462 bp, which was the PEDV M gene, whereas the cells inoculated with PEDV-CV777 did not. To verify the infection of FHs 74 Int cells by PEDV-LJX and PEDV-CV777, the expression of the PEDV N protein in the cells was detected via a red fluorescence assay. The results showed that PEDV-LJX could infect FHs 74 Int cells, with a peak infection at 24h followed by a decrease (Figure 2A,C). In contrast, PEDV-CV777 could not infect FHs 74 Int cells (Figure 2B).

### 3.2. Characterization of the S Gene of PEDV LJX after Inoculation in FHs 74 Int Cells

To examine if genetic changes occurred in the S gene of PEDV during passage in FHs 74 Int cells, the complete S gene in FHs 74 Int cells after PEDV-LJX infection was amplified and sequenced. The sequenced S gene was 4149 nucleotides long, encoding a protein of 1382 amino acids (aa). Compared with the S gene from the cell-cultured PEDV-SX strain, the PEDV-LJX S gene showed 100% homology, and the relative homologies with strains CV777, DR13, and JS2018 were 96.9%, 98.4%, and 99.7%, respectively. Compared with the classical attenuated vaccine strain PEDV CV777, there existed 53 aa mutations in the S gene of the PEDV-LJX strain (Figure 3).

Phylogenetic analysis was carried out based on the S gene of the PEDV-LJX from infected FHs 74 Int cells and other PEDV strains obtained from GenBank. The result showed that the PEDV strain LJX from FHs 74 Int cells was closely related to the Chinese strains AH-M and SX and was distantly related to Chinese strains PEDV JA-A and AJ1102 as well as the HN-VN strain isolated from Vietnam (Figure 4).

### 3.3. PEDV Particle Could Be Detected in FHs 74 Int Cells

To further confirm that the PEDV-LJX strain can infect FHS 74 Int cells, PEDV-LJX infected FHs 74 Int cells were observed by electron microscopy. The results showed the presence of PEDV particles in FHs 74 Int cells at 24 h post-infection, and some viral particles were observed within the vacuole (Figure 5).

### 3.4. Proliferative Rule of PEDV in IPEC-J2 and FHs 74 Int Cells

To investigate proliferative rule of PEDV in porcine intestinal epithelial cells (IPEC-J2) and human intestinal epithelial cells (FHs 74 Int), we detected M gene mRNA transcripts and N gene protein expression by qRT-PCR and Western blotting, respectively, in infected IPEC-J2 cells and FHs 74 Int cells. The data demonstrated that the expression level of the PEDV N protein was the highest at 12 h post-infection (hpi) in IPEC-J2 cells and then declined to the lowest level at 48 h (Figure 6A,C). The proliferation rule of PEDV in FHs 74 Int cells was consistent with that of IPEC-J2 (Figure 6B,D). Thereafter, we used the TCID_50_ method to measure changes in virus titer after PEDV infection with FHS 74 Int cells over 12 to 48 h. As shown in Figure 6E, the virus titer showed a continuous decreasing trend after 12 hpi.

Then, FHs 74 Int cells were infected with PEDV-LJX strains at an MOI of 0.1. The viral titers (TCID_50_) at 12, 24, 36, and 48 hpi were measured. The results revealed that PEDV began to proliferate from 0 to 12 hpi. The virus titer reached a peak at 12 hpi, after which its proliferation slowed down from 12 h to 48 h. The viral RNA copy number was assessed based on the standard curve for the M gene of PEDV using qPCR (Figure 7A,B). The results showed that the proliferative rule of PEDV-LJX between IPEC-J2 cells and FHs 74 Int cells was similar, and the amount of PEDV M RNA reached a peak at 12 hpi and then gradually decrease from 24 to 48 hpi (Figure 7C,D). The above data suggested that the proliferation rule of PEDV-LJX infection in IPEC-J2 cells and FHs 74 Int cells was consistent.

## 4. Discussion

Concurrently with the global expansion of humans and domestic mammals, various coronaviruses have emerged as a result of cross-species transmission among humans and domestic and wild animals. Coronaviruses can infect humans and many different animal species, and most human viral pathogens originated in animals and arose through cross-species transmission. Both SARS-CoV and MERS-CoV are zoonotic pathogens of animal origin that are thought to have been transmitted to humans from natural hosts (possibly from bats) via some intermediate mammalian reservoir [21]. Phylogenetic analysis of the full genome sequence of SARS-CoV-2 indicated that, much like SARS-CoV, the new human coronavirus is likely to have originated in a bat host [22]. Viruses with the spike of SHC014-CoV, a SARS-like virus currently circulating in the Chinese horseshoe bat population, are able to replicate efficiently in human primary airway cells [23]. In recent years, an increasing number of studies have shown that porcine coronaviruses can infect human cells across species; SADS-CoV can infect human hepatoma cells (Huh-7, HepG2/C3A), human embryonic kidney cells (293T), human lung cancer cells (A549), human cervical adenocarcinoma cells (HeLa), and human intestinal cells (HRT-18, Caco-2) [24,25]. In addition, transient expression of APN makes human Hela cells susceptible to PDCoV infection [26]. Live virus infection experiments have shown that PEDV can effectively infect human hepatocellular carcinoma cells (Huh-7) and human embryonic lung fibroblasts (MRC-5) and that it can replicate in human embryonic kidney cells (HEK 293) [27,28]. PEDV is known to infect human lung cells, hepatocytes, and kidney cells; however, it has not been reported whether PEDV infects human small intestinal epithelial cells. Therefore, it is necessary to study the susceptibility of human small intestinal epithelial cells to PEDV, which will help us to further understand the host and tropism of PEDV and will provide a theoretical reference for assessing the potential risk of PEDV to humans, which has public health significance.

There are numerous studies showing cross-immunity between the N and S proteins of different coronavirus species. For example, SARS-CoV antibodies in the sera of recovered SARS patients can bind to other Beta coronaviruses (MERS-CoV and hCoV-OC43) [29]. Cross-immunity is possible even in different genera, and studies have shown that polyclonal antibodies to antigenic group I coronaviruses, including human coronavirus 229E (hCoV-229E), feline infectious peritonitis virus (FIPV), and porcine transmissible gastroenteritis virus (TGEV), react strongly with the SARS-CoV antigen [30]. As a common human coronavirus, recombinant proteins at amino acids 59–377 and 59–271 of the HCoV-NL63 N protein can bind to sheep anti-hCoV-229E antiserum [31]. Moreover, Fouchier et al. showed by evolutionary tree analysis that hCoV-NL63 is on the same branch as PEDV and hCoV-229E, and they have high amino acid sequence homology with each other [32]. Binding of PEDV antigens to human coronavirus antibodies is not excluded.

In the present study, we determined that human small intestinal cells (FHs 74 Int) were susceptible to PEDV infection. We assessed PEDV growth kinetics through a one-step growth-curve experiment in IPEC-J2 and FHs 74 Int cell lines, the supernatants of which were taken at set times and titrated on Vero cells. The results indicated that these cell lines are indeed permissive for PEDV infection. Interestingly, PEDV-CV777 could not infect FHs 74 Int cells, and the underlying mechanism needs to be further investigated. In addition, we do not know through which receptors PEDV infects human small intestinal epithelial cells or whether PEDV can infect human small intestinal epithelial cells using the known human coronavirus receptors (e.g., ACE2, DPP4, and APN). Answering these questions will provide insights into the cross-species transmission properties of PEDV.

## 5. Conclusions

In this study, we demonstrated that PEDV can infect FHs 74 Int cells, suggesting that PEDV has the potential to spread across species. Under the conditions of the COVID-19 pandemic, the possibility of cross-species transmission of PEDV also implies the possibility of SARS-CoV-2 recombination, which warrants further vigilance and investigation.

## Figures and Tables

**Figure 1 viruses-15-00956-f001:**
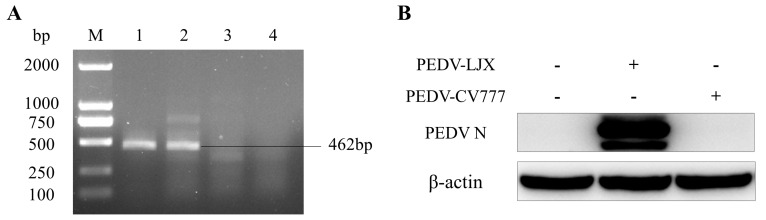
(**A**) M gene expression after infection of FHs 74 Int cells with different strains of PEDV. M: DL 2000 Marker; 1: PEDV positive control; 2: PEDV-LJX + FHs 74 Int; 3: PEDV-CV777 + FHs 74 Int; 4: FHs 74 Int. (**B**) N protein levels after infection with FHs 74 Int cells with different strains of PEDV. PEDV, porcine epidemic diarrhea virus; FHs 74 Int cells, human small intestine epithelial cells.

**Figure 2 viruses-15-00956-f002:**
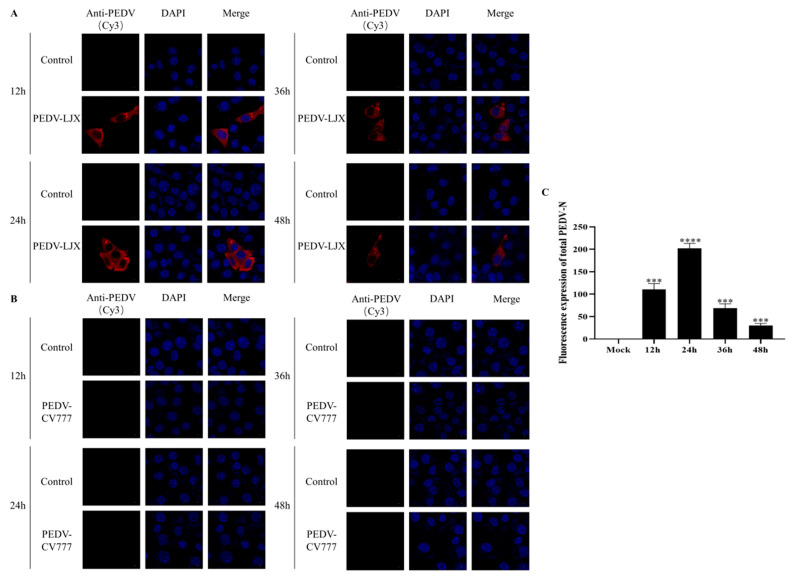
(**A**) Immunofluorescence results of PEDV-LJX infection of FHs 74 Int cells (Cy3: PEDV N protein, DAPI: nucleus). (**B**) Immunofluorescence results of PEDV-CV777 strain infection with FHs 74 Int cells. (**C**) Quantitative analysis of immunofluorescence for the PEDV-N protein after infection with PEDV-LJX. Cy3, cyanine 3; DAPI, 4′,6-diamidino-2-phenylindole. (*** *p*-value < 0.001; **** *p*-value < 0.0001).

**Figure 3 viruses-15-00956-f003:**
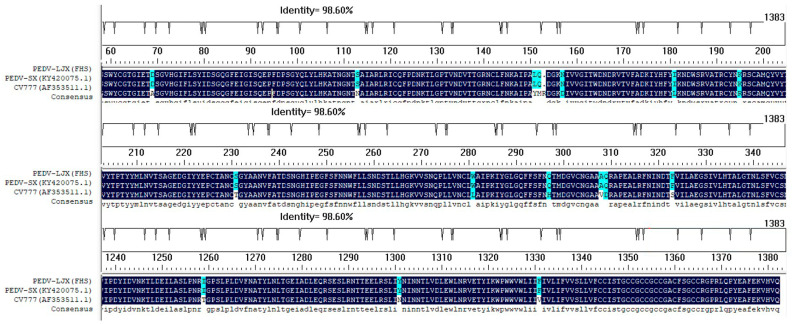
Alignment of S protein sequences in different PEDV strains. Reference S protein sequences obtained from GenBank are indicated by their strain names and GenBank accession numbers. The mutated amino acids in the S protein in PEDV-LJX are indicated in blue.

**Figure 4 viruses-15-00956-f004:**
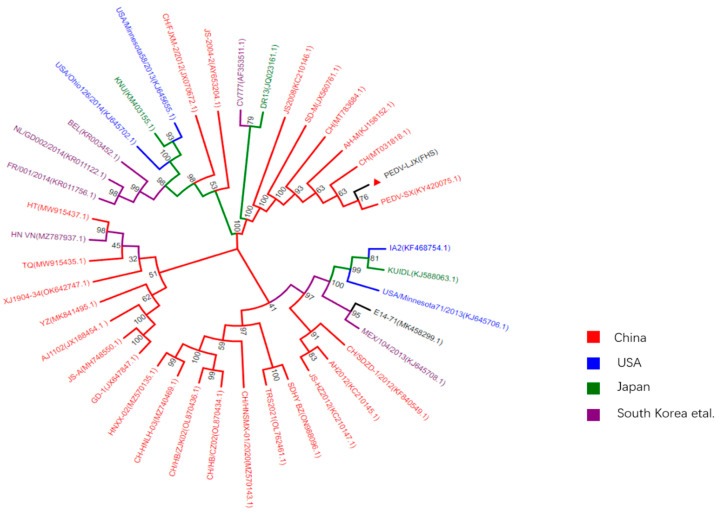
Phylogenetic analysis based on the S genes from different PEDV strains. The phylogenetic tree was constructed from the aligned nucleotide sequences using the neighbor-joining method with MEGA11 and EvolView software. Reference sequences obtained from GenBank are indicated by their strain names and GenBank accession numbers. The S gene of the PEDV-LJX strain infecting FHs 74 Int cells is indicated by the red triangle.

**Figure 5 viruses-15-00956-f005:**
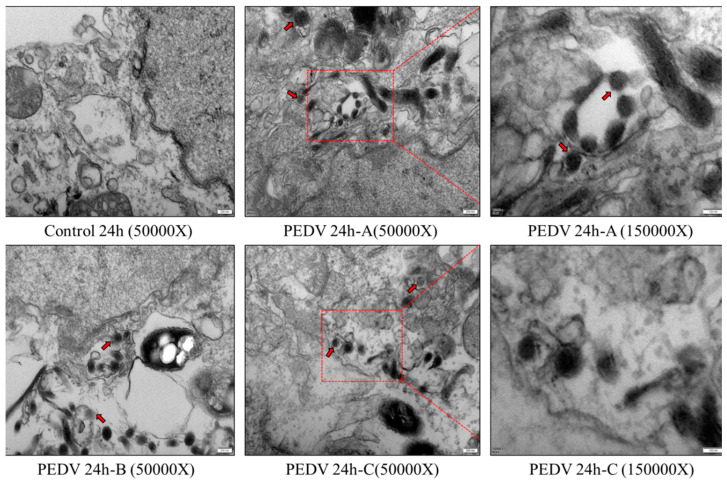
Electron microscopy results of PEDV-infected FHs 74 Int cells. No virus particles were observed in the cell control group, and virus particles could be observed in FHs 74 Int cells at 24 h post-infection. Magnification 50,000× and 150,000×. The red box represents the enlarged part of the figure on the right; The arrows show the virus, the PEDV virion.

**Figure 6 viruses-15-00956-f006:**
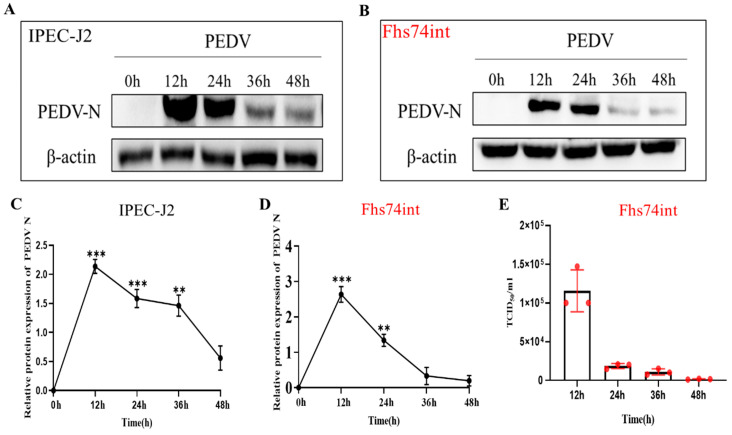
Expression levels of M genes at different time points in PEDV-infected IPEC-J2 cells and FHs 74 Int cells. (**A**) Amplification of the PEDV M gene product using fluorescent primers. (**B**) Absolute quantitative standard curve of PEDV M gene expression. (**C**) Expression of the M gene in PEDV-infected FHs 74 Int cells. (**D**) Expression of the M gene in PEDV-infected IPEC-J2 cells. (**E**) Changes of viral titer after infection with FHS 74 Int cells 12, 24, 36 and 48 by the PEDV-LJX strain. (** *p*-value < 0.01; *** *p*-value < 0.001).

**Figure 7 viruses-15-00956-f007:**
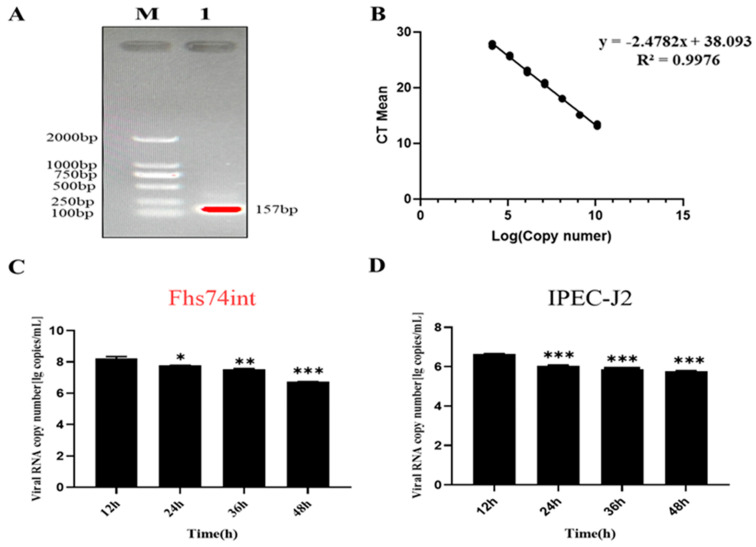
Expression levels of M genes at different time points in PEDV-infected IPEC-J2 cells and FHs 74 Int cells. (**A**) Amplification of PEDV M gene product by fluorescent primers. (**B**) Absolute quantitative standard curve of PEDV M gene. (**C**) Expression of M gene in PEDV−infected FHs 74 Int cells. (**D**) Expression of M gene in PEDV-infected IPEC-J2 cells. (* *p*-value < 0.05; ** *p*-value < 0.01; *** *p*-value < 0.001).

**Table 1 viruses-15-00956-t001:** Primer sequences for the PEDV M gene.

Gene	Sequence (5′–3′)	bp
*PEDV M*	F: GTCTAACGGTTCTATTCCC	462
	R: TTATAGCCCTCTACAAGC	

**Table 2 viruses-15-00956-t002:** Primer sequences for the amplification of the full-length PEDV S gene.

Gene	Sequence (5′–3′)
*PEDV S1*	F: TGCTAGTGCGTAATAATGACACC
*PEDV S3*	R: GTTGGCAGACTTTGAGACA

**Table 3 viruses-15-00956-t003:** Primer sequences for the PEDV M gene.

Gene	Sequence (5′–3′)	bp
*PEDV M*	F: AGGTTGCTACTGGCGTACAG	157
	R: GAGTAGTCGCCGTGTTTGGA	

## Data Availability

The raw data supporting the conclusions of this article will be made available by the authors without undue reservation.

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
