# Peer review of "Porcine Epidemic Diarrhea Virus Replication in Human Intestinal Cells Reveals Potential Susceptibility to Cross-Species Infection"

_viruses, 2023, doi:10.3390/v15040956_

Round 1
Reviewer 1 Report
In this manuscript, Niu et al. found that the PEDV LJX strain can infect human small intestinal epithelial cells (FHs 74 Int cells), and they also found that the PEDV CV777 strain cannot cause infection. Their findings have important implications for the cross-species spread of Coronaviruses and their possible recombination. But there are some areas that should be modified accordingly to clarify and add scientific significance to the manuscript.
Major points:
1. The use of the full name and abbreviation of cells in the manuscript is slightly confusing, some places are abbreviations, but some places are full names, which is more obvious in the use of human small intestinal epithelial cells (FHs 74 Int cells).
2. In 2.5, where did the plasmids used to construct the standard curve during absolute real-time PCR come from? Not mentioned in the manuscript.
3. In Result 3.1, what are the PCR primers for the PEDV M gene?
4. The results in 3.2 are not reflected in the Materials and Methods.
5. In lines 203-204, "To further verify whether the PEDV-LJX strain could infect human small intestinal cells," the expression seems slightly inappropriate, PEDV LJX has been determined to infect FHS 74 Int cells in the previous experiment, and the use of "whether" here is incorrect
6. Fig. 6E is not stated in the text
7. English of this manuscript needs to be improved.
Minor points:
1. In line 16, when “Porcine intestinal epithelial cells” mentioned for the first time in the manuscript, it’s abbreviations (IPEC-J2) should be marked thereafter
2. In section 2.1, the source of the PEDV cv777 strain is not clearly explained.
3. What is the "NHE3 gene" in line 125, "The NHE3 gene was..."?
4. The title of 3.1 is vague.
5. Line 157-159, “To further investigate...in FHs 74 Int cell lines" differs from the order in Fig. 1 of the result.
6. In line 215, the abbreviation of "human intestinal epithelial cells" is incorrect.
Author Response
Responses to Reviewer 1:
Major points:
- The use of the full name and abbreviation of cells in the manuscript is slightly confusing, some places are abbreviations, but some places are full names, which is more obvious in the use of human small intestinal epithelial cells (FHs 74 Int cells).
- Thank you for the reminder. We have rechecked the full manuscript and corrected the errors that existed.
- In 2.5, where did the plasmids used to construct the standard curve during absolute real-time PCR come from? Not mentioned in the manuscript.
-Thank you for this comment. The plasmids used for absolute PCR were stored in our laboratory and we have added with this information to the revised manuscript (lines 158-159). Thank you again.
- In Result 3.1, what are the PCR primers for the PEDV M gene?
- We apologize for our negligence; we have added the relevant test methods and primer sequences to section 2.2. Thank you for pointing out our problem.
- The results in 3.2 are not reflected in the Materials and Methods.
- Thank you for this comment. We have added the test methods to the revised manuscript in lines 140-151.
- In lines 203-204, "To further verify whether the PEDV-LJX strain could infect human small intestinal cells," the expression seems slightly inappropriate, PEDV LJX has been determined to infect FHS 74 Int cells in the previous experiment, and the use of "whether" here is incorrect
- Thank you for pointing this out. We have changed this to "To further confirm that the PEDV-LJX strain can infect FHS 74 Int cells".
- Fig. 6E is not stated in the text
- Thank you for your comments. We have referred to Fig.6E in lines 258-261 in the revised manuscript.
- English of this manuscript needs to be improved.
-We apologize for the poor use of language in the original manuscript. With the assistance of a native English speaker with an appropriate research background, the language of manuscript has been effectively improved.
Minor points:
- In line 16, when “Porcine intestinal epithelial cells” mentioned for the first time in the manuscript, it’s abbreviations (IPEC-J2) should be marked thereafter
-Thank you for this comment. We have modified the manuscript accordingly (line 16).
- In section 2.1, the source of the PEDV cv777 strain is not clearly explained.
-We apologize for this omission. The PEDV CV777 strain was saved by our laboratory and we have added this source to 2.1 (lines 78-79).
- What is the "NHE3 gene" in line 125, "The NHE3 gene was..."?
-Thank you for this comment. We have modified the relevant text (line 164).
- The title of 3.1 is vague.
- Thank you for your advice. We have revised the title of 3.1 to "PEDV LJX can infect FHs 74 Int cells, but PEDV CV777 cannot".
- Line 157-159, “To further investigate...in FHs 74 Int cell lines" differs from the order in Fig. 1 of the result.
- Thanks for pointing this out. To match the order of the manuscript description, we have reformatted Fig.1, as shown in the manuscript (line 207).
- In line 215, the abbreviation of "human intestinal epithelial cells" is incorrect.
- Thank you. We have corrected to the errors in the manuscript (line 253).
Reviewer 2 Report
The article aimed to prove PEDV could infect and propagate in human intestinal epithelial cells, which is an interesting topic and significant for potentially zoonostic control. However, the innovation of the study is limited and some of the results was not convincing enough. Besides, the article was not well written and the language needs extensive editing.
1. Innovation of the study is limited, as other researchers have proved PEDV could infect human cells. It is not so significant the cells are of epithelial or other kinds.
2. From the results, it is possible that PEDV infected the FHs74 cells only transiently. The cells might not support a full replication of the virus, as the author did not provide result of passages of the virus in the cells.
3. in result 3.3 and figure 5, how the author could prove these dark dots in the pictures were PEDV particles?
4. Figure legends were incomplete, for instance Fig1A, Fig1B, Fig2A.
Author Response
- Innovation of the study is limited, as other researchers have proved PEDV could infect human cells. It is not so significant the cells are of epithelial or other kinds.
- Thank you for your comment. Currently, various coronaviruses have emerged as a result of cross‑species transmission among humans, and domestic and animals. Intestinal diseases, e.g., diarrhea, are important symptoms caused by coronaviruses, such as SARS-CoV-2. PEDV causes acute diarrhea vomiting, dehydration, and high mortality in neonatal piglets. Small intestinal epithelial cells are the target cells for PEDV infection, and determining whether PEDV can infect human small intestinal epithelial cells is important to reveal the potential risk of PEDV cross-species infection in humans. The present study addressed this problem. We demonstrated that PEDV is capable of infecting human small intestinal epithelial cells, thus posing a risk of cross-species transmission.
- From the results, it is possible that PEDV infected the FHs74 cells only transiently. The cells might not support a full replication of the virus, as the author did not provide result of passages of the virus in the cells.
- Thank you for this question. So far, we have carried out passaging for more than 20 generations, the result showed that PEDV still successfully infects FHs74 cells.
- In result 3.3 and figure 5, how the author could prove these dark dots in the pictures were PEDV particles?
- Thank you for this advice. Electron microscopy is an important method to identify viruses. In this study, we also used RT-PCR, western blotting, IFA and other methods to determine that PEDV-LJX can infect FHS 74 Int cells. Electron microscopy demonstrated the presence of virions in cells/vesicles. In future research, we will use immunoelectron microscopy for confirmation.
- Figure legends were incomplete, for instance Fig1A, Fig1B, Fig2A.
- Thank you for this comment. We have updated the legends.
Round 2
Reviewer 2 Report
The article was improved considerably after authors' revision. Two minor revision suggestions:
1. Line 263. the virus titer reached a peak at 12hp2, not correct.
2. The are two figure7 legends and in the legends the second C should be D.
Author Response
- Line 263. the virus titer reached a peak at 12hp2, not correct.
- Thank you for the correct. We have corrected the error on line 263.
- The are two figure7 legends and in the legends the second C should be D.
-Thank you for this comment. The first Figure 7 (line 272) should be Figure 6; in addition, all legends have been checked and corrected for errors (line 275 and 280). Thank you again.